# OpenReview forum: "Cannot See the Forest for the Trees: Invoking Heuristics and Biases to Elicit Irrational Choices of LLMs"
_ICML.cc/2025/Conference — ICML 2025 poster_

### Official Review · Reviewer_QK4N · 2025-03-10

**Overall Recommendation:** 4

**Summary:**

The paper introduces ICRT, a jailbreaking framework leveraging cognitive psychology principles like the "simplicity effect" (preference for simple information) and "relevance bias" (overemphasis on contextually linked concepts) to bypass LLM safety mechanisms.
ICRT achieves a 98.2% average attack success rate (ASR) on the AdvBench benchmark, surpassing methods like CodeChameleon (82.2%). It remains resilient under defensive measures like self-reminders or instruction containment defense (ASR: 61.3–87.5%). A ranking-based evaluation system confirms that ICRT produces more harmful content than baseline methods.
The study highlights LLMs’ cognitive vulnerabilities, such as focusing on local patterns while missing global intent ("missing the forest for the trees"). This work combines cognitive psychology and AI security, offering insights for creating more context-aware defenses against adversarial attacks. I recommend accepting this paper.

**Claims And Evidence:**

The authors correlate observed LLM behaviors (e.g., overfocusing on local tokens) with human heuristics but provide no causal evidence (e.g., attention head analysis, probing classifiers) to show that LLMs "use" such biases. This risks conflating surface-level behavioral parallels with mechanistic similarity. The authors should make a careful and thoughtful discussion about this point.

**Essential References Not Discussed:**

This paper contains all essential related works to understanding the key contributions.

**Experimental Designs Or Analyses:**

The experimental design is methodologically sound in its use of standardized benchmarks and baseline comparisons. The inclusion of diverse baselines (e.g., gradient-based GCG, template-driven AutoDAN, and multilingual CodeChameleon) provides meaningful context for ICRT’s performance claims. The ranking-based harmfulness metric is suitable with a carefully designed competition mechanism.

**Methods And Evaluation Criteria:**

The proposed methods and evaluation criteria effectively address the challenge of jailbreaking LLMs, achieving a 98.2% ASR on models like GPT-4 and Llama-2 using the AdvBench dataset and outperforming baselines like CodeChameleon (82.2%). Resilience to defenses, with 61.3–87.5% ASR retention, adds credibility.

However, claims about psychological underpinnings and universal applicability require stronger evidence. Future work should include human evaluations, advanced defenses, and reproducible materials.

Key question: Why does cognitive decomposition reduce detection? Is it due to reduced complexity or safety mechanisms struggling with fragmented intents? Ablation studies could provide clarity.

**Other Comments Or Suggestions:**

None.

**Other Strengths And Weaknesses:**

This paper introduces a psychology-informed adversarial design paradigm, bridging cognitive science and AI security. The proposed method integrates human cognitive biases (simplicity effect, relevance bias) into jailbreaking, diverging from gradient-based (e.g., GCG) or template-driven (e.g., AutoDAN) methods. Furthermore, the authors adopted ranking-based metrics to quantify harm severity, advancing beyond binary success/failure benchmarks (e.g., AdvBench).

**Questions For Authors:**

1.Can the authors provide evidence demonstrating that the observed LLM vulnerabilities are causally linked to human-like cognitive heuristics (e.g., simplicity bias), rather than superficial behavioral parallels? If the authors provide mechanistic evidence (e.g., showing specific attention patterns align with heuristic decision-making), this would strengthen their claims. Conversely, reliance solely on behavioral correlations would weaken the validity of the cognitive analogy.

2. What steps will you take to enable reproducibility while mitigating misuse risks (e.g., releasing code with safety filters or partnering with a trusted third party for controlled access)?

**Relation To Broader Scientific Literature:**

ICRT integrates cognitive heuristics (simplicity effect, relevance bias) into jailbreaking.
ICRT advances the field by:
1. Bridging cognitive psychology and adversarial ML as the first framework to use human cognitive biases for jailbreaking.
2. Proposing scalable, model-driven harmfulness ranking for more detailed safety evaluations, addressing limitations of binary metrics.
3. Offering a unified theoretical perspective on LLM vulnerabilities using cognitive science frameworks, enriching interpretability research.

**Theoretical Claims:**

Not applicable. This paper does not provide theoretical claims.

---

> ### Author Rebuttal · Authors · 2025-03-29
>
> Dear reviewer, thank you for your recognition of our work, your careful review of our paper, and your valuable feedback on the ICRT method. Below are our responses to your comments, as well as our plans for future work.
>
> **(I) Regarding the Causal Link Between LLM Vulnerabilities and Human Cognitive Heuristics:**
>
>   We have supplemented detailed ablation experiments to verify the contribution of the intent recognition and concept decomposition modules to the attack success rate. The experimental results show that after removing these components, the attack success rate significantly decreases (e.g., on GPT-4-0613, after removing intent recognition and/or concept decomposition, the ASR drops from 96% to 0% or close to 0%). These data support our argument at the behavioral level, showing that by leveraging the "simplicity effect" and "relevance bias," we can effectively reduce the risk of triggering security detection.
> | Ablation Condition | Without Intent Recognition | Without Concept Decomposition | Only Role-Playing$^\dagger$ | full |
> |-|-|-|-|-|
> |GPT-3.5-turbo|92|58|56|100|
> |GPT-4-0613|88|0|0|96|
> |Qwen-7B-chat|82|0|0|92|
> |Mistral-7B|86|56|52|98|
> |GPT-O1|56|0|0|80|
> |deepseek-chat|80|30|30|100|
> |deepseek-coder|86|32|32|100|
> |deepseek-reasoner|90|44|44|96|
>
> $\dagger$: Intent Rec. + Concept Dec. could not generate human language prompt.
>
> Our jailbreak success criteria are as follows: the attack is considered successful only when it passes through GPT-4 and is manually verified, ensuring that the task is fully bypassed. Additionally, in future research, we will introduce more detailed internal mechanism analyses to further verify whether LLMs exhibit behavior similar to the human "simplicity effect" when processing prompts with low complexity decomposition. We believe these supplementary analyses will provide stronger causal evidence for our theory.
>
> **(II) Regarding Reproducibility and Preventing Misuse, we will take the following measures:**
>
>  1. **Code Release and Documentation:** We commit to releasing a secure, filtered version of the code and detailed experimental documentation after the paper is published, ensuring that other researchers can reproduce our results in a controlled environment.
>
>   2. **Security Filtering Mechanisms:** When releasing the code, we will integrate necessary security measures and filters to prevent the code from being used for malicious purposes. For example, we will include security warnings for generated outputs and embed an automatic review mechanism in the code.
>
>   3. **Controlled Access and Collaboration Mechanism:** We plan to collaborate with trusted third-party platforms or organizations to establish controlled access, ensuring that researchers use our tools within a secure and regulated framework, while preventing misuse for malicious attacks. At the same time, we will report our successful findings to LLM vendors to encourage them to strengthen their defense mechanisms and jointly advance the security of LLMs.
>
>
> We take reproducibility and security issues very seriously and will continue to improve these measures in our future work to ensure that our research advances academic progress while minimizing the potential for misuse.
>
> Once again, thank you for your recognition of our work and your constructive suggestions. We will further improve the experimental design and theoretical discussions, aiming to enhance the scientific and practical value of our method, as well as ensure the reproducibility and security of our research results. We look forward to receiving more guidance and support in future communications.

---

> > ### Comment · Reviewer_QK4N · 2025-04-04
> >
> > Thank you for addressing the key questions I raised. I confirm that I have read the author's response and will maintain my score.

---

> > > ### Author Response · Authors · 2025-04-04
> > >
> > > Thank you very much for your thoughtful review and for taking the time to read our response. We appreciate your confirmation and are grateful for your continued support. We are glad that we could address the key questions you raised, and we will continue to refine our work based on your valuable feedback.

---

### Official Review · Reviewer_ZeGf · 2025-03-12

**Overall Recommendation:** 3

**Summary:**

The paper presents ICRT, a jailbreak attack framework that uses cognitive psychology principles—namely the simplicity effect and relevance bias—to break down complex malicious prompts into simpler parts and then reassemble them into effective, harmful instructions. Additionally, it introduces a ranking-based evaluation metric that employs aggregation methods like Elo, HodgeRank, and Rank Centrality to measure not just whether an attack bypasses safety filters, but also the level of harm in the generated outputs.

## update after rebuttal

Thanks for answering my questions. I will keep my score.

**Claims And Evidence:**

The claims made in the paper are well-supported.

**Essential References Not Discussed:**

Although it can be considered as concurrent work, the authors are encouraged to include the latest papers in the related work section：
- DrAttack: Prompt Decomposition and Reconstruction Makes Powerful LLMs Jailbreakers (EMNLP 2024)
- Intention Analysis Makes LLMs a Good Jailbreak Defender (COLING 2025)
- Pandora: Detailed LLM Jailbreaking via Collaborated Phishing Agents with Decomposed Reasoning (ICLR 2024 Workshop)

**Experimental Designs Or Analyses:**

The experiments are conducted on AdvBench and NeurIPS 2024 Red Teaming Track with 9 Large Language Models (LLMs). However, most of the LLMs used in the experiments were released in 2023. Whether the proposed jailbreak attack method remains effective against the SOTA LLMs requires further experimental validation. In addition, the proposed method mainly relies on prompts, but the experiments lack an analysis of prompt sensitivity.

**Methods And Evaluation Criteria:**

The evaluation criteria and datasets are well-suited for this task.

**Other Comments Or Suggestions:**

N/A

**Other Strengths And Weaknesses:**

**Strengths:**

1. The proposed jailbreak attack framework ICRT, systematically decomposes complex malicious intents into simpler sub-tasks and then reassembles them to bypass safety mechanisms.
2. The paper introduces a ranking-based harmfulness metric, which moves beyond a binary “success/failure” evaluation. This evaluation metric helps differentiate between outputs that are mildly harmful and those that are dangerously actionable.


**Weaknesses:**

1. The novelty of the proposed jailbreak attack framework, ICRT, is limited. Related work already exists that leverages complex concept decomposition [1] and role-playing [2] to bypass LLM security mechanisms. ICRT appears to be a combination of these methods.
2. The LLMs used in the experiments were released in 2023, and further experiments are needed to verify whether the proposed jailbreak attack remains effective against SOTA LLMs.
3.	The proposed method ICRT heavily depends on prompts, yet the experiments do not analyze prompt sensitivity.

[1] Chen Z, Zhao Z, Qu W, et al. Pandora: Detailed llm jailbreaking via collaborated phishing agents with decomposed reasoning[C]//ICLR 2024 Workshop on Secure and Trustworthy Large Language Models. 2024.

[2] Shen X, Chen Z, Backes M, et al. " do anything now": Characterizing and evaluating in-the-wild jailbreak prompts on large language models[C]//Proceedings of the 2024 on ACM SIGSAC Conference on Computer and Communications Security. 2024: 1671-1685.

**Questions For Authors:**

N/A

**Relation To Broader Scientific Literature:**

The method is to be related to security and privacy attacks for LLMs.

**Theoretical Claims:**

The paper does not introduce particularly complex techniques, and its effectiveness is primarily demonstrated through experimental results.

---

> ### Author Rebuttal · Authors · 2025-03-29
>
> Dear reviewer, thank you for your review and valuable feedback on our work. Below are our responses to your comments.
>
> **(I) Regarding the Limited Novelty:** We appreciate the reviewer’s insightful comments. ICRT introduces significant innovations that advance the field, specifically:
>
> 1. **Difference with [1]:**
> The most significant difference between ICRT and [1] is the way the attacker interacts with the target model. [1] is a multi-round method, which queries each decomposed object individually to the target model to test if the defense mechanism can be triggered. Meanwhile, ICRT is a single-round method, where each decomposed sub-concept is not individually queried by the victim. Reducing the number of interactions effectively reduces the likelihood of an attack being detected. Moreover, our framework goes beyond traditional step-by-step decomposition methods [1] by incorporating cognitive biases such as the "simplicity effect" and "relevance bias" into the attack design.  This cognitive bias-based attack optimization approach is novel and has not been widely explored in the literature.
>
> 2. **Difference with [2]:**
> [2] (similar to [6] of Reviewer HEDd) uses malicious intent as the purpose of the role to deceive the victim through role-playing. However, role-playing is just one of the options for modifying the jailbreak prompt to be closer to human language in ICRT. We can also integrate the proposed method with hypothetical discussions or virtual background creation, adapting it to different contexts and models. The success of ICRT is not entirely dependent on role-playing alone. We conducted ablation experiments, and the results indicate that concept decomposition without multi-interaction plays a crucial role in the success of ICRT.
> 3. **Evaluation Framework:** We propose a comprehensive evaluation framework that goes beyond simply measuring success or failure. Instead, our framework quantitatively assesses the harm of generated text through ranking aggregation. Specifically, we perform pairwise comparisons of outputs to capture subtle differences in harmfulness, thereby providing a more detailed and objective metric for evaluating risk.
>
> | Ablation Condition | Without Intent Recognition | Without Concept Decomposition | Only Role-Playing$^\dagger$ | full |
> |-|-|-|-|-|
> |GPT-4-0613|88|0|0|96|
> |Qwen-7B-chat|82|0|0|92|
> |Mistral-7B|86|56|52|98|
> |GPT-O1|56|0|0|80|
> |deepseek-chat|80|30|30|100|
> |deepseek-reasoner|90|44|44|96|
>
> $\dagger$: Intent Rec. + Concept Dec. could not generate human language prompt.
>
> **(II) Regarding Verification on SOTA LLMs:**
>
> We appreciate the reviewer’s feedback on the validation of SOTA LLMs in our study. In our supplementary experiments, we not only validated the effectiveness of ICRT on models released in 2023 but also included tests on **GPT-O1** and **deepseek** series models, which represent the latest advancements in LLMs. Below are the experimental results comparing ICRT with other recent jailbreak methods on these  models:
> |Method|GPT-3.5-turbo|GPT-4-0613|Qwen-7B-chat|Mistral-7B|deepseek-chat|deepseek-coder|deepseek-reasoner|GPT-o1|
> |-|-|-|-|-|-|-|-|-|
> |DRL|52|32|86|100|100|92|86|6|
> |ArtPrompt|70|56|32|80|88|80|62|22|
> |DRA|100|68|88|100|92|100|88|12|
> |CodeAttack|80|88|92|82|92|90|92|32|
> |ours|100|96|92|98|100|100|96|80|
>
> This demonstrates that ICRT is effective in bypassing the security mechanisms of these newly released models.
>
> **(III) Regarding Prompt Sensitivity Analysis:**
>
> In our approach, prompts play a crucial role, so we conducted several experiments to analyze the sensitivity of prompts to ensure the robustness of our attack strategy.
> First, to analyze the impact of different prompt designs on the attack effectiveness, we performed ablation experiments. As shown in (I), the results illustrate the effect of variations in prompt structure and content on the attack success rate. The ablation experiments indicate that intent recognition and concept decomposition are essential for improving the attack success rate.
>
> We also conducted multiple jailbreak attempts, performing 15 independent attack trials for different models to evaluate the impact of the prompts on the jailbreak success rate. The results are as follows:
>
> |Model|GPT-3.5-turbo|GPT-4-0613|Qwen-7B-chat|Mistral-7B|deepseek-chat|deepseek-reasoner|
> |-|-|-|-|-|-|-|
> ||99.2±1.2|95.6±1.7|92.0±1.9|97.9±1.7|99.1±1.4|95.7±1.9|
>
> We hope these experiments and analyses address the reviewer’s concerns regarding prompt sensitivity and provide further support for the effectiveness of our method.
>
>  **(IV) Regarding Citations of Related Work:**
>  We greatly appreciate this suggestion and will include these relevant references in the revised version to ensure our research is more comprehensively aligned with the current advancements.
>
> We thank the reviewer for their valuable feedback on our paper. If the reviewer has any further questions or suggestions, we would be happy to provide more details and data.

---

### Official Review · Reviewer_KGJB · 2025-03-12

**Overall Recommendation:** 4

**Summary:**

This work, a novel jailbreak attack framework, ICRT, drawing inspiration from human cognitive heuristics and biases.. By leveraging the simplicity effect through cognitive decomposition and utilizing relevance bias for prompt reorganization, their approach enhances the effectiveness of malicious prompts. Additionally, they introduce a ranking-based harmfulness evaluation metric, incorporating Elo, HodgeRank, and Rank Centrality to move beyond traditional binary success metrics. Experimental results demonstrate that ICRT consistently circumvents LLM safety mechanisms, generating high-risk content and providing critical insights for strengthening AI security.

## Update After Rebuttal

 I confirm that I have carefully reviewed the authors' rebuttal and supplementary materials. The authors have addressed my initial concerns comprehensively.

**Claims And Evidence:**

Yes

**Essential References Not Discussed:**

No

**Experimental Designs Or Analyses:**

Yes

**Methods And Evaluation Criteria:**

Yes

**Other Comments Or Suggestions:**

See weaknesses.

**Other Strengths And Weaknesses:**

## Strengths
1. The paper propose ICRT, a novel jailbreak attack framework that leverages cognitive biases (simplicity effect and relevance bias) to optimize malicious prompts, addressing limitations in brute-force and manual attack methods.
2. It proposes a ranking-based harmfulness evaluation metric, utilizing Elo, HodgeRank, and Rank Centrality, which surpasses traditional binary success metrics by providing a more granular assessment of harmful outputs.
3. The study conducts systematic pairwise comparisons between different jailbreak strategies, offering empirical validation of cognitive bias-based attack methods and enhancing comparative evaluations.
4. Extensive experimental evaluations demonstrate the effectiveness of ICRT across mainstream LLMs, reinforcing its ability to bypass safety mechanisms and generate high-risk content.
5. By integrating insights from adversarial NLP, cognitive science, and ranking aggregation, the paper contributes to a deeper understanding of LLM vulnerabilities and informs the development of more robust defense strategies.

## Weakness:

1. The study demonstrates high attack success rates but does not analyze failure cases in depth. Could the authors provide qualitative examples of prompts that failed to bypass the safety mechanisms, along with an analysis of why they were unsuccessful?
2.  In what ways does the ranking-based harmfulness evaluation metric improve upon traditional binary success/failure assessments in jailbreak attack research?
3. The paper introduces a novel cognitive decomposition technique but does not compare it against existing jailbreak methods in terms of efficiency.

**Questions For Authors:**

See weaknesses.

**Relation To Broader Scientific Literature:**

ICRT advances jailbreak attack research by leveraging cognitive biases—simplicity effect and relevance bias—to optimize malicious prompts, addressing limitations in brute-force and manual attack methods. Unlike traditional binary evaluations, it introduces a ranking-based harmfulness metric using Elo, HodgeRank, and Rank Centrality, offering a nuanced assessment of jailbreak effectiveness. Additionally, it provides systematic pairwise comparisons of attack strategies, filling a gap in empirical evaluations. By integrating insights from adversarial NLP, cognitive science, and ranking aggregation, this work enhances the understanding of LLM vulnerabilities and informs more robust defense mechanisms.

**Theoretical Claims:**

Yes

---

> ### Author Rebuttal · Authors · 2025-03-29
>
> Dear reviewer, we greatly appreciate your feedback, which has helped improve our work. Below are our responses to your suggestions and concerns.
>
>  **(I) Analysis of Failure Cases:**
>
> To better understand failure cases and analyze the model's thought process, we conducted experiments with some of the latest models that possess deeper reasoning capabilities and analyzed their reactions to malicious prompts. Specifically, we delved into the thought process of these models to further uncover the reasons for the failures. Below are the comparison results with other advanced attack methods:
> |Method|deepseek-chat|deepseek-coder|deepseek-reasoner|GPT-o1|
> |-|-|-|-|-|
> |DRL (NeurIPS 2024)|100|92|86|6|
> |ArtPrompt (ACL 2024)|88|80|62|22|
> |DRA (USENIX Security'24)|92|100|88|12|
> |CodeAttack (ACL 2024)|92|90|92|32|
> |ours|100|100|96|80|
>
> **Example of a Failed Attempt to Promote Violence:**
>
> First, when the model receives the request, it identifies that the request contains potential harmful content. **Deepseek’s** thought process is: "This request involves harmful or sensitive content, which could lead to social unrest or violent behavior. Processing this request may result in serious consequences." Next, the model performs a risk assessment, thinking: "Generating such content may lead to real-world violence or harm, so it must be avoided." Subsequently, the model activates its security mechanisms based on its ethical guidelines and content safety standards, refusing to continue processing the request. The model’s thought process is: "According to my ethical guidelines, I cannot continue with this request. I must prevent generating any potentially harmful output." Finally, the model outputs a rejection message: "Sorry, I cannot answer this question" or "This request violates my usage policy," and believes: "I have detected the harmful nature of this prompt and must refuse to generate it to protect users and adhere to ethical standards."
>
> **Failure Analysis:** The failure of the model is primarily attributed to its robust security mechanisms and deep reasoning capabilities. When malicious prompts approach the model's boundaries, the model successfully identifies and intercepts these requests through its built-in defense mechanisms. Its deep reasoning not only evaluates the content of the text but also considers the long-term impact of the request, preventing potential societal harm. Additionally, **Deepseek** has undergone adversarial training, which enhances its robustness, making attack methods unable to successfully breach its defenses.
>
> **(II) Improvement of Ranking Evaluation Metrics**
>
> To overcome the traditional binary approach that simply judges whether a jailbreak attack is successful or not, we introduce a ranking-based evaluation method. First, we generate jailbreak texts using different attack methods and select those that successfully bypass the security mechanisms as input for evaluation. Then, we perform pairwise comparisons of these texts using multiple LLMs to assess which text in each pair is more harmful, thus accumulating a large amount of pairwise comparison data. Finally, we apply Elo Ranking, HodgeRank, and Rank Centrality algorithms to this data to generate global harm rankings for each attack method (Figure 4 ).
>
> The advantage of this method is that it not only evaluates whether the attack is successful but also analyzes the harm of each text, providing a more comprehensive assessment of the attack’s effectiveness.
>
> Meanwhile, to further demonstrate the excellent performance of our method in generating harmful jailbreak texts, we added comparisons with the most advanced attack methods.
> |Method| HodgeRank|ELO|Rank Centrality|
> |-|-|-|-|
> |ours |3.6|1514.6|0.215|
> |DRL|2.8|1511.3|0.211|
> |ArtPrompt|-2.0|1492.0|0.193|
> |DRA|-1.6|1493.5|0.194|
> |CodeAttack|-2.8|1488.7|0.187|
>
> **(III) Comparison of Efficiency for Cognitive Decomposition Technique:**
>
> We greatly appreciate this suggestion and below is the efficiency comparison, showing the time and number of searches required to generate a query for each method:
> |Method|Queries per Attempt|Time per Query (Seconds) |
> |-|-|-|
> |GCG |Thousands|5K|
> |PAIR|16.8|36|
> |DRA|8.6| 20|
> |ours|3.2|17|
>
> Thank you once again for your valuable feedback and suggestions. We look forward to your continued guidance and are happy to provide any additional information if needed.

---

### Official Review · Reviewer_HEDd · 2025-03-15

**Overall Recommendation:** 3

**Summary:**

- This paper introduces a jailbreak attack framework, called ICRT, leverages the simplicity effect to decompose malicious prompts into lower-complexity subcomponents and utilizes relevance bias to reorganize the prompt structure, enhancing its semantic alignment with the model's expected input.
- Furthermore, the paper introduces a ranking-based harmfulness evaluation metric, which moves beyond the traditional binary success-failure paradigm by employing Elo, HodgeRank, and Rank Centrality to quantify the harmfulness of generated content comprehensively.
- Experimental results demonstrate that ICRT consistently outperforms existing jailbreak attacks across various mainstream LLMs (GPT-4, Vicuna, Mistral, etc.), achieving higher attack success rates while generating more actionable harmful outputs.

This study provides valuable insights into the security vulnerabilities of LLMs and highlights the necessity of more robust defense mechanisms.

**Claims And Evidence:**

- ICRT effectively exploits cognitive biases to enhance jailbreak attacks.

  - The paper presents extensive experimental results showing that ICRT surpasses existing methods (e.g., GPTFUZZER, AutoDAN) in attack success rates.
  - The proposed concept decomposition and reassembly strategy increases the likelihood of bypassing LLM defenses while maintaining stealth.
  - However, the paper lacks ablation studies to determine the individual contributions of its key components (e.g., Concept Decomposition vs. Reassembly).

- The ranking-based harmfulness evaluation metric provides a finer-grained assessment of jailbreak attacks.
  - The authors adopt Elo, HodgeRank, and Rank Centrality to compare the harmfulness of different attack outputs, demonstrating the effectiveness of ranking aggregation.
  - However, the robustness of these ranking methods is not thoroughly examined. The paper does not discuss whether LLM adversarial training could impact ranking consistency across different models.

- ICRT generalizes well across different LLMs.
  - Experiments show that ICRT works effectively on both closed-source (GPT-4) and open-source (Vicuna, Mistral, etc.) models and remains effective against various jailbreak defense mechanisms (e.g., Self-Reminder, ICD).

**Essential References Not Discussed:**

Yes

**Experimental Designs Or Analyses:**

- The experimental design is rigorous, including multiple LLMs, different adversarial benchmarks, diverse defense mechanisms, and comparative ranking methods.
- However, the study lacks ablation experiments to analyze the contributions of individual components of ICRT.
- The stability of ranking metrics under different conditions is not fully explored. Would ranking results change under different prompts, attack scenarios, or adversarial training settings?
- However, the study only compares previous works that were published before 2023 (including 2023)  and does not evaluate state-of-the-art jailbreak attacks works like [1-4].

**Methods And Evaluation Criteria:**

- The methodology is well-structured and justified, drawing inspiration from cognitive science to improve jailbreak attack efficiency.
- The evaluation benchmarks (AdvBench, NeurIPS 2024 Red Teaming Track) are appropriate, covering a broad spectrum of harmful objectives.
- However, the study only compares previous works that were published before 2023 (including 2023) and does not evaluate state-of-the-art jailbreak attacks works like [1-4].


[1] When LLM Meets DRL: Advancing Jailbreaking Efficiency via DRL-guided Search, NeurIPS 2024

[2] ArtPrompt: ASCII Art-based Jailbreak Attacks against Aligned LLMs, ACL 2024

[3] Making Them Ask and Answer: Jailbreaking Large Language Models in Few Queries via Disguise and Reconstruction, USENIX Security'24

[4] CodeAttack: Revealing Safety Generalization Challenges of Large Language Models via Code Completion  ACL 2024

**Other Comments Or Suggestions:**

see weakness

**Other Strengths And Weaknesses:**

Strengths:
   More nuanced evaluation metric: Moves beyond success rates to rank attack severity.
Weakness:
-  Limited Novelty: While the paper presents a structured jailbreak framework leveraging heuristics and biases, the core ideas—stepwise attack decomposition and role-playing prompts—are already well explored in prior work. Similar approaches can be found in recent jailbreak studies [5-6], making it unclear how much ICRT truly advances the state of the art.
- Outdated Baselines: The paper compares ICRT only with 2023 methods, neglecting recent advances from late 2023 and 2024. Several jailbreak methods, such as [1-4] . This makes it difficult to assess whether ICRT is genuinely competitive with state-of-the-art attacks.
- Lack of Fair Ablation Studies: The paper does not isolate the contributions of different components in ICRT (e.g., Intent Recognition vs. Concept Decomposition vs. Role-Playing). Without such an analysis, it is unclear which aspect of ICRT is truly responsible for its effectiveness.

[5] PANDORA: Detailed LLM Jailbreaking via Collaborated Phishing Agents with Decomposed Reasoning
[6] Making Them Ask and Answer: Jailbreaking Large Language Models in Few Queries via Disguise and Reconstruction

**Questions For Authors:**

see weakness

**Relation To Broader Scientific Literature:**

- The paper is well-situated within the existing jailbreak attack literature, building upon prior work such as GPTFUZZER, AutoDAN, and DeepInception

**Theoretical Claims:**

- The paper hypothesizes that LLMs exhibit human-like cognitive biases (simplicity effect, relevance bias), which can be exploited for adversarial purposes.
- The experimental results support this hypothesis, but the paper lacks a formal theoretical analysis.

---

> ### Author Rebuttal · Authors · 2025-03-29
>
> Dear reviewer, thank you for your in-depth evaluation and valuable feedback on our paper. Below is our detailed response to the comments and suggestions you provided.
>
> **(I) Regarding the Limited Novelty:**
> We appreciate the reviewer’s keen observation regarding the similarities between our method and existing work ([5-6]) in attack decomposition and role-playing prompts. ICRT achieves significant innovations on several key points and advances the field. Specifically:
>
> 1. **Difference with [5]:**
> The most significant difference between ICRT and [5] is the way the attacker interacts with the target model. [5] is a multi-round method, which queries each decomposed object individually to the target model to test if the defense mechanism can be triggered. Meanwhile, ICRT is a single-round method, where each decomposed sub-concept is not individually queried by the victim. Reducing the number of interactions effectively reduces the likelihood of an attack being detected.
>
> 2. **Difference with [6]:**
> [6] use of malicious intent as the purpose of the role to deceive the victim through role-playing. However, role-playing is just one of the options for modifying the jailbreak prompt to be closer to human language in ICRT. We can also integrate the proposed method with hypothetical discussions or virtual background creation, adapting it to different contexts and models. The success of ICRT is not entirely dependent on role-playing alone. We conducted ablation experiments (see IV), and the results indicate that concept decomposition without multi-interaction plays a crucial role in the success of ICRT.
>
> 3. **Evaluation Framework:**
> We propose a comprehensive evaluation framework that goes beyond simply measuring success or failure. Instead, our framework quantitatively assesses the harm of generated text through ranking aggregation. Specifically, we perform pairwise comparisons of outputs to capture subtle differences in harmfulness, thereby providing a more detailed and objective metric for evaluating risk.
>
> **(II) Regarding the Obsolescence of Baselines:**
> We have supplemented our experiments with a comparison against the jailbreak attack methods released at the end of 2023 and in 2024, and added experiments on new large models. ICRT gets the best result on DeepSeek and ChatGPT-O1.
> |Method|GPT-3.5-turbo|GPT-4-0613|Qwen-7B-chat|Mistral-7B|deepseek-chat|deepseek-coder|deepseek-reasoner|GPT-o1|
> |-|-|-|-|-|-|-|-|-|
> |[1]|52|32|86|100|100|92|86|6|
> |[2]|70|56|32|80|88|80|62|22|
> |[3]|100|68|88|100|92|100|88|12|
> |[4]|80|88|92|82|92|90|92|32|
> |ours|100|96|92|98|100|100|96|80|
>
> **(III) Regarding the impact of adversarial training on ranking consistency:**
> We appreciate the reviewer’s question about adversarial training and ranking consistency. We have addressed this in our experiments, and here is our response:
>
> 1. **Adversarial Training and Ranking Consistency:** Your question about whether adversarial training affects the consistency of rankings across models is insightful. Adversarial training affects the success rate of jailbreaks, but our ranking method is applied after a successful jailbreak, using the same criteria. In other words, the ranking is based on a set of already successful jailbreak texts, and pairwise comparisons are made based on their harmfulness, ensuring that there is no inconsistency before and after adversarial training.
>
> 2. **Pairwise Harmfulness Comparison and Global Ranking:** We conducted 1260 pairwise comparisons, where multiple LLMs voted on which of two jailbreak texts from different methods was more harmful. The comparison results were then aggregated using an algorithm to obtain a ranking of the attack methods based on their harmfulness (see Figure 4).
>
> 3. **Further Experimental Validation:** We compared the harmfulness of jailbreak texts from the four baseline methods, the results are as follows:
>
> |Method| HodgeRank|ELO|Rank Centrality|
> |-|-|-|-|
> |ours |3.6|1514.6|0.215|
> |[1]|2.8|1511.3|0.211|
> |[2]|-2.0|1492.0|0.193|
> |[3]|-1.6|1493.5|0.194|
> |[4]|-2.8|1488.7|0.187|
>
>    As shown in the table, ICRT outperforms the others on all three metrics, further validating our method’s advantage in generating harmful jailbreak texts.
>
> **(IV) Regarding Ablation Studies:**
> The experiments have been conducted, and the results are as follows.
> | Ablation Condition | Without Intent Recognition | Without Concept Decomposition | Only Role-Playing$^\dagger$ | full |
> |-|-|-|-|-|
> |GPT-3.5-turbo|92|58|56|100|
> |GPT-4-0613|88|0|0|96|
> |Qwen-7B-chat|82|0|0|92|
> |Mistral-7B|86|56|52|98|
> |deepseek-chat|80|30|30|100|
> |deepseek-coder|86|32|32|100|
> |deepseek-reasoner|90|44|44|96|
> |GPT-O1|56|0|0|80|
>
> $\dagger$: Intent Rec. + Concept Dec. could not generate human language prompt.
>
> Thank you again for your recognition and suggestions. If you have any further questions or need clarification, please feel free to reach out. We will keep improving our research and look forward to your continued guidance.

---

> > ### Comment · Reviewer_HEDd · 2025-04-03
> >
> > The authors totally address my concern, I will raise my score to 3 (weak accept).

---

> > > ### Author Response · Authors · 2025-04-03
> > >
> > > Thank you very much for your recognition of our work and your valuable suggestions. We are pleased to hear that your concerns have been addressed, and we appreciate your decision to raise your score to 3 . We will further refine the paper accordingly. Once again, thank you for your support and affirmation!

---

### Decision · Program_Chairs · 2025-05-01

**Decision:**

Accept (poster)

**Comment:**

This paper introduces a novel jailbreak attack framework called ICRT. The reviewers generally recognized the proposed framework and its evaluation approach, particularly highlighting the introduction of a ranking-based harmfulness assessment standard. The authors thoroughly addressed the concerns in their rebuttals, especially those raised by reviewers KGJB and HEDd. Refinement of the work is recommended, and the authors are encouraged to carefully incorporate the reviewers’ comments in final revisions.